# Validation of Earphone-Type Sensors for Non-Invasive and Objective Swallowing Function Assessment

**DOI:** 10.3390/s22145176

**Published:** 2022-07-11

**Authors:** Takuto Yoshimoto, Kazuhiro Taniguchi, Satoshi Kurose, Yutaka Kimura

**Affiliations:** 1Department of Health Science, Kansai Medical University, Hirakata 573-1010, Osaka, Japan; kurosesa@hirakata.kmu.ac.jp (S.K.); kimuray@hirakata.kmu.ac.jp (Y.K.); 2Department of Rehabilitation, Hirakata Kohsai Hospital, Hirakata 573-0153, Osaka, Japan; 3Department of Aesthetic Design and Technology, Yasuda Women’s University, Hiroshima 731-0153, Hiroshima, Japan; taniguchi-k@yasuda-u.ac.jp

**Keywords:** swallowing, sensor, soft palate, noninvasive

## Abstract

Standard methods for swallowing function evaluation are videofluoroscopy (VF) and videoendoscopy, which are invasive and have test limitations. We examined the use of an earphone-type sensor to noninvasively evaluate soft palate movement in comparison with VF. Six healthy adults wore earphone sensors and swallowed barium water while being filmed by VF. A light-emitting diode at the sensor tip irradiated infrared light into the ear canal, and a phototransistor received the reflected light to detect changes in ear canal movement, including that of the eardrum. Considering that the soft palate movement corresponded to the sensor waveform, a Bland–Altman analysis was performed on the difference in time recorded by each measurement method. The average difference between the time taken from the most downward retracted position before swallowing to the most upward position during swallowing of the soft palate in VF was −0.01 ± 0.14 s. The Bland–Altman analysis showed no fixed or proportional error. The minimal detectable change was 0.28 s. This is the first noninvasive swallowing function evaluation through the ear canal. The earphone-type sensor enabled us to measure the time from the most retracted to the most raised soft palate position during swallowing and validated this method for clinical application.

## 1. Background

Pneumonia is one of the leading causes of hospitalization and death in older adults worldwide. It has been reported that approximately 80% of older adults hospitalized for pneumonia have “aspiration pneumonia” and that this proportion increases with age [1]. Aspiration pneumonia is mainly caused by invasion of oral bacteria from the trachea into the lungs together with food and saliva due to deterioration of swallowing function.

Standard methods that have already been established to evaluate swallowing function include videofluoroscopy (VF) and videoendoscopy (VE). VF is the gold standard for evaluating swallowing function, wherein a subject swallows a liquid or a bolus containing a contrast agent while being observed under X-ray fluoroscopy. This technique can help in observing the dynamics of each organ involved in swallowing beyond the oral cavity and determining whether aspiration has occurred. In VE, a fiber scope is inserted from the nasal cavity to observe the movement of various organs involved in swallowing and identify the extent of food residues [1,2].

Both of these methods allow the easy evaluation of the entire swallowing process and provide useful information; however, they require expensive testing equipment and environments as well as the technical proficiency of the physicians conducting the test and cannot be easily performed in individuals at their home or in patients under home care [1]. Furthermore, VF is associated with the risk of radiation exposure, contrast medium aspiration, and contrast agent allergies [3,4,5]. In many cases, patients report pain and discomfort when the fiber scope is inserted for VE. These two methods are also characterized by a testing environment that is markedly different from the usual eating/dietary environment of individuals.

Conversely, methods for evaluating swallowing function that do not require specific equipment include questionnaires and the repetitive saliva swallowing test (RSST) [6,7]. In the RSST, the examiner palpates the thyroid cartilage of the subject and checks how many times the subject is able to swallow voluntarily during a 30 s period to assess the risk of aspiration. It is a simple test that does not require any equipment, but there are often clinical cases of patients with reduced cognitive function, such as in dementia, who have difficulty in understanding questions or instructions, thereby affecting the correct evaluation based on the attention and motivation of patients. Dementia is reported to be a predictor of the onset of aspiration pneumonia in older adults [8,9,10], and while the number of patients with dementia is increasing rapidly, there is an urgent need to establish a method of evaluation that can be applied even to patients with dementia with aspiration pneumonia.

In recent years, studies have focused on non-invasive and simple methods for evaluating swallowing function, such as evaluation of the tongue, hyoid bone movement, and tongue pressure using ultrasonography, electromyography, and pressure sensors, as well as the assessment of mastication and swallowing using microphones [11,12,13,14]. All these methods are non-invasive and useful means for obtaining information, but we cannot rule out the possibility that the method of applying an ultrasonic probe to the mandible while swallowing or attaching a sensor to the skin near the oral cavity or larynx may interfere with the movement of various organs involved in swallowing or may cause discomfort. Additionally, when sensors are attached to the skin, errors may occur depending on the thickness of subcutaneous fat and the degree of excess skin and this may pose difficulties in patients with skin fragility [15]. Furthermore, since examination circumstances that vary from routine environments can be the cause of confusion or stress for patients with dementia, it would be ideal for the evaluation of swallowing function to be performed in daily eating/dietary environments [16,17]. Although there have been many studies exploring simple methods to evaluate tongue movement, tongue pressure, and hyoid bone movement [11,12,13,14], a method that can easily evaluate the soft palate movement, despite its important role in preventing regurgitation of food bolus in the nasal canal while swallowing and generating pressure for the transport of the food bolus, remains to be established [18].

To address this limitation, we have developed an earphone-type sensor to objectively measure swallowing function (Figure 1). Most people have used earphones in their lives, and these do not interfere with chewing and swallowing motions. Therefore, compared with existing evaluation methods, we believe that earphone-type sensors will allow for easier assessment of swallowing under conditions that are closer to swallowing motions that normally occur during routine meals.

This study aimed to verify the validity of the earphone-type sensor by simultaneously recording measurements with VF, the gold standard for evaluating swallowing function, and drawing comparisons between both these approaches to determine whether the movement of the soft palate can be evaluated.

## 2. Methods

### 2.1. Subjects

Six healthy adults were enrolled and assigned numbers from subject No. 1 to No. 6. We recruited volunteers who were healthy adults aged 20–60 years and excluded individuals presenting with organic defects in the oral cavity, pharynx, and larynx (prostheses were allowed). The six subjects were No. 1 (26 years old, male), No. 2 (56 years old, female), No. 3 (38 years old, female), No. 4 (41 years old, female), No. 5 (29 years old, male), and No. 6 (34 years old, male).

This study was conducted with the approval of the Kansai Medical University (Approval No.: 2020303) and after briefing all subjects about the purpose and content of the research thoroughly and obtaining their written consent.

### 2.2. Videofluoroscopy Swallowing Test

Two examiners were involved in testing: one individual sat in the operating room and recorded the video footage on the external monitor (Examiner α) and the other (Examiner β) operated the earphone-type sensor and voice-recorder in the X-ray room and also provided the samples to the subjects. Subjects sat on a chair in the X-ray room and maintained the sitting position by adjusting the height of the seat so that the knees and hip joint were 90° apart and both feet were flat on the floor with a plantar dorsiflexion of the ankle of 0°. Thereafter, an earphone-type sensor was inserted into their left ear and a voice recording microphone (throat microphone, Nanzu Radio, Shimoda-shi, Japan) was attached to the larynx. The sample contrast agent (barium sulfate powder 97.5% “HORII” Horii Pharmaceutical Ind., Osaka, Japan) was prepared by adjusting barium sulfate to a concentration of ≥30–40% by weight, based on the standard testing method of the Japanese Society of Dysphagia Rehabilitation [19]. The volume of contrast agent was measured by a syringe, such that the volume swallowed each time was 3 mL. Examiner β provided a cup of the solution to each subject for each trial. Subjects were instructed to face the front, keep the head and neck in the intermediate position, and avoid moving the head and neck as much as possible. A single trial was defined from the point in which Examiner β provided the subject with a container holding 3 mL of barium water to the point where the recipient subject swallows the barium water at once; a total of five trials were performed. We performed five trials per subject with continuous video recording, considering the level of fatigue of each subject and radiation exposure. Filming was conducted by a physician and a radiotechnologist, who took lateral images using an X-ray TV device (DREX-FR80/J1, Canon, Tokyo, Japan), and the second Examiner α recorded the footage on the external monitor. The footage on the external monitor was continuously recorded through all five trials for each subject. For each trial of each subject, a video editing software (Video Pad v 10.41, NCH Software, Canberra, Australia) and video analysis software (Image J, National Institutes of Health, Bethesda, MD, USA) were used to detect and record the time the soft palate takes to reach its lowest and most retracted position (VA), the time the soft palate takes to reach its highest and most advanced position (VB), and the time the soft palate takes to go down again (VC), based on the recorded data.

### 2.3. Earphone-Type Sensor

The earphone-type sensor is very easy to mount, and various information can be provided simply by inserting the earphones into the ear canal. It is a highly reliable apparatus that can allow measurements of mastication count [20], respiratory rate [21], occlusal force [22], mealtime [23], and tongue movement [24]. The earphone-type sensor developed for this study (Figure 1A) had a longer elastic region than conventional earphone-type sensors that measure mastication rate, respiratory rate, occlusal force, mealtime, or tongue movement. Specifically, while the length of elastic material is approximately 5 mm in conventional sensors, the one in the sensor used for taking measurements of the swallowing motion was 15 mm. To create the elastic material, commercially available earplugs were altered (EP4 SONIC DEFENDERS: PLUS Filtered Flanged Earplugs (SureFire LLC., Fountain Valley, CA, USA). By increasing the length of the elastic material, the photosensor could be inserted deep into the ear canal, making it easier to measure changes in the shape of the eardrum and the ear canal near the eardrum. Since the elastic material modifies its shape to fit the shape of the ear canal, the sensor can accommodate individual differences in the shape of the ear canal. The eardrum continues to the pharynx through the Eustachian tube (Figure 1B) and may reflect the movement of the soft palate involved in the opening of the Eustachian tube.

The earphone-type sensor was the same shape as earplug-style earphones. Its built-in optical distance sensor “QRE1113” (Fairchild Semiconductor International Inc., Sunnyvale, CA, USA) used an infrared LED and phototransistor to penetrate the inner ear canal with infrared light. When the ear canal is irradiated with infrared light using an LED (wavelength: 960 nm) and the irradiated area is moved, the way the light emitted from the light-emitting part is reflected changes. When the phototransistor receives the reflected light, it detects the movement of the ear canal, including the eardrum. The light energy of the detected light is converted and amplified into electrical energy, and the signal is converted into a digital signal by an analog–digital (hereinafter referred to as AD) conversion circuit. The movement of the ear canal during swallowing could be measured using the data thereby obtained (Figure 1C). Figure 1D shows the electronic circuits surrounding the lightwave distance sensor. As Figure 1D depicts, when the distance (d) between the tympanic membrane and the optical distance sensor decreased, the amount of light reflected from the tympanic membrane increased, along with the output voltage. Likewise, an increase in distance (d) led to a decrease in reflected light and output voltage.

The earphone-type sensor was connected via cable to the measuring device. The size of measuring device was 110 × 75 × 25 mm and weighed 115 g; it was small enough that it would be able to sit on a dining table without getting in the way. The measuring device supplied a voltage of DC3.3 V to the earphone-type sensor, and the sensor’s output data was detected by the measuring device’s offset voltage regulator. The offset voltage levels measured from the sensor were adjusted to central values after AD (analog–digital) conversion of the signal received by the offset voltage regulator of the measuring device. The adjustment was then set to the center value of the AD convertible range = 3.3 V (power supply voltage of the AD converter)/2 = 1.65 V. This adjustment of the offset voltage was necessary to compensate for differences in this parameter caused by individual differences in the shape of the ear canal. The value (waveform) measured by the sensor was the amplitude based on the offset voltage. Because the offset voltage was fixed by the amplifier, only the amplitude was enlarged in the signal after adjustment of offset voltage. Using a knob (variable resistor) on the instrument, the amplification level could be raised up by to 40×. After amplification, the analog signal was converted to a digital signal by an AD converter with a sampling frequency of 250 Hz and a resolution of 10 bits. Finally, the converted digital signal was transmitted to a tablet (ASUS Nexus 7, Bluetooth 3.0, ASUS, Taipei, Taiwan) by a transmitter (Bluetooth 2.1) [20].

Measurement results can be shown and recorded on a tablet terminal at a remote location using Bluetooth. In this study, the earphone-type sensors continuously recorded data from the first to fifth trial for each subject. The subjects placed and held the sample in their oral cavity and pressed a switch immediately before swallowing to detect the timing of swallowing, then pressed the switch again immediately after completing the swallowing. The data between the two pressings of the switch were considered true values during the swallowing motion. From the data recorded on the tablet terminal, the data between the switch pressing was extracted and converted into a waveform with the AD conversion value of voltage on the vertical axis and time on the horizontal axis using spreadsheet software (Excel, Microsoft, Redmond, WA, USA). The points where the waveform was lowest (SA), highest (SB), and lowering again (SC) were detected, and the timing of emergence of these points were recorded (Figure 2).

The results of measuring the movement of the soft palate with the earphone-type sensor are shown. The X-axis shows the time (seconds) and the Y-axis shows an AD conversion of the measured values of the sensor, using a resolution of 10 bits and a sampling frequency of 250 Hz. VA: time the soft palate reaches its lowest and most retracted position; VB: time the soft palate reaches its highest and most advanced position; VC: time the soft palate goes down again; SA: time the waveform was at its lowest point; SB: time, the waveform was at its highest point; SC: time the waveform decreased; DA: difference in the time of emergence of VA and SA; DB: difference in the time of emergence between VB and SB; DC: difference in the time of emergence of VC between SC; VI: time from VA to VB; VII: time from VB to VC; VIII: time from VA to VC; SI: time from SA to SB; SII: time from SB to SC; SIII: time from SA to SC. Difference between TI:VI and SI; difference between TII:VII and SII; and difference between TIII:VIII and SIII.

### 2.4. Data Processing

Since Examiners α and β started recording with the VF and the sensor, respectively, the start time of the recording differed. This time lag was calculated based on the voice recorded at the same time, and the time of emergence of SA to SC was corrected and used for analysis (Figure 3). In addition, waveforms that could not be interpreted due to improper sensor mounting, switch pressing mistakes or noise, were excluded from the subsequent analysis. Assuming that the time of emergence the of VA and SA, VB and SB, and VC and SC corresponded, we defined DA as the difference in time of emergence of VA and SA, DB as the difference between VB and SB, DC as the difference between VC and SC, VI as the time from VA to VB, VII as the time from VB to VC, VIII as the time from VA to VC, SI as the time from SA to SB, SII as the time from SB to SC, SIII as the time from SA to SC, TI as the difference between VI and SI, TII as the difference between VII and SII, and TII as the difference between VIII and SIII (Figure 3). The average of all trials was defined by the weighted average of the number of trials for each subject.

### 2.5. Statistical Analysis

Measurements are expressed in terms of average ± standard deviation. Bland–Altman plots were drawn using the time of TI, TII, and TIII [25]. The Bland–Altman analysis is used to study the difference in paired measurements taken at the same time by two different methods. The X-axis shows the average values of measurement pairs, and the Y-axis shows the differences between the methods, thus allowing to visually and statistically determine the presence or absence of systematic errors (fixed errors or proportional errors) in the measurements [26]. This study required five repeated measurements for each subject, subject-specific errors, and measurement method errors were associated with the measurements. Therefore, based on the method of Bland et al., analysis of variance was performed with each subject as a factorial, and the upper and lower ends of the 95% match limit were obtained considering repeated measurements [27]. If a one-sample t-test did not include 0 in the 95% confidence interval of the average difference of measurements and the measurements were all distributed in either the positive or negative direction, we regarded this as a sign of fixed error being present; whereas the proportional error was considered to be present when the regression formula resulting from the Bland–Altman plot regression analysis was deemed significant [26]. If there was neither a fixed error nor a proportional error, the minimum detectable error (MDC) was determined, and the measurement method was deemed appropriate for clinical use. Although a tolerable range of degree of agreement has not been defined [28], of the methods proposed as the standard of compatibility, if the number of measurements with a relative error of ≤30% or was ≥75% of the total number of measurements, we defined it as compatible [29]. A statistical analysis software (R version 4.1.2, R Development Core Team, Vienna, Austria) and a spreadsheet software (Excel, Microsoft, Washington, DC, USA) were used for statistical analyses.

## 3. Results

A total of 30 swallowing data trials were performed for the six subjects and, of these, only 27 data trials were accepted for analysis. The first trial for subject No. 2 was excluded due to poor sensor mounting, the first trial for subject No. 4 was excluded because it was indiscernible, and the fourth trial of subject No. 6 was excluded for a switch-pressing error. Figure 4A shows the waveform resulting from poor sensor mounting, while Figure 5 shows the waveform from data deemed indiscernible. The average swallowing times indicated by the subjects pressing switches, from subjects No. 1 to No. 6, were 1.38, 3.10, 2.60, 2.32, 5.49, and 3.03 s, respectively.

Table 1 shows the differences (DA, DB, and DC) between the times of emergence of soft palate movement in VF footage (VA, VB, and VC) and the times of emergence of waveform movement based on the earphone-type sensor (SA, SB, and SC) in terms of average ± standard deviation (SD). Figure 5 illustrates the total sensor error for VF for each subject from the results in Table 1. As shown in Table 1 and Figure 5, the smallest errors in the VF and sensor readings were observed in No. 3, with errors of DA: 0.09 ± 0.07 s, DB: 0.13 ± 0.07 s and DC: 0.12 ± 0.04 s. The largest errors were seen for subject No. 2, with DA: 0.50 ± 0.15 s, DB: 0.54 ± 0.12 s and DC: 0.83 ± 0.30 s.

Table 2 shows the time taken (V1) for the soft palate to go move from the lowest and most retracted position (VA) to the highest and most advanced position (VB), the time taken (VII) from VB to the soft palate that goes down again (VC), the time taken from VA to VC (VII), the time taken (SI) from the lowest point of the sensor wave (SA) to the highest point of the sensor wave (SB), the time taken (SII) from SB to the wave that goes down again (SC) and the time taken from SA to SC (SIII), in terms of average ± standard deviation.

Figure 6 shows the Bland–Altman plot created for VI and SI, VI and SII, and VI and SIII based on the results in Table 2. Figure 6A shows VI and SI, Figure 6B shows VIII and SII, and Figure 6C shows VIIII and SIII. The plots were color-coded for each subject. For TI, the average difference was −0.01 ± 0.14 s and the 95% LOA was −0.28 to 0.28 s. The 95% confidence interval (CI) of the one-sample t-test was −0.06 to 0.05 s and the correlation coefficient of the Bland–Altman plot was −0.13 (*p* > 0.05), which meant there was no fixed error or proportional error. The MDC was 0.28 s. Of the 27 measurements, 17 (63%) had relative errors below ±30%. The average values were 0.38 ± 0.14 s for VI and 0.38 ± 0.10 s for SI. For TII, the mean difference was −0.33 ± 0.23 s and the 95% LOA was −0.79 to 0.13 s. The 95% CI of the one-sample t-test was −0.42 to −0.24 s, indicating fixed error, while the correlation coefficient of the Bland–Altman plot was −0.01 (*p* > 0.05), which meant there was no proportional error. Of the 27 measurements, 4 (approximately 15%) had relative errors below ±30%. For TIII, the average difference was −0.34 ± 0.31 s and the 95% LOA was −0.97 to 0.28 s. The 95% CI of the one-sample t-test was −0.46 to −0.21 s and indicated a fixed error, while the correlation coefficient of the Bland–Altman plot was −0.13 (*p* > 0.05) and meant that there was no proportional error. Of the 27 measurements, 11 (approximately 40%) had relative errors below ±30%.

## 4. Discussion

In this study, we examined the validity of the earphone-type sensor by simultaneously recording measurements with VF and compared these approaches to determine whether the movement of the soft palate can be evaluated. Of the total of 30 measurements obtained from six subjects, we were able to collect 27 swallowing data trials. To the best of our knowledge, the method used in this study is the first approach to evaluate swallowing function noninvasively from the ear canal, thus providing novel findings.

Regarding the mechanism by which the soft palate movement could be measured with the earphone-type sensor, it is possible that the reflected light received by the sensor was able to better reflect the movement of the eardrum due to the more slender shape of the sensor tip and vicinity to the eardrum. The distance from the ear canal to the eardrum has been reported to be approximately 25 mm for adults [30], and the length of elastic material of sensor was approximately 15 mm, indicating that it was close to the eardrum. The eardrum is approximately 0.1 mm thick, and it moves due to changes in pressure from the Eustachian tube, which is the cavity that connects the pharynx and the tympanic cavity and is primarily dilated by the tensor veli palatine muscle, while the soft palate is primarily elevated by the levator veli palatine muscle, which has a stop in the Eustachian tube [31]. Since the series of fluid movements from the elevation of the soft palate to the dilation of the Eustachian tube is performed by coordinated movement of these palate muscles, the pressure changes of the Eustachian tube due to the soft palate movement is reflected in the movement of the eardrum, and our sensor was able to detect these movements.

In this study, the validity of the earphone-type sensor was analyzed by comparing the time of emergence of the movement and the time required for the movement, assuming that the sensor waveform corresponded to the time of the soft palate movement (Figure 2). We applied the Bland–Altman analysis used for method-comparison studies to examine TI, TII, and TIII (Figure 6). No plot was created with the DA, DB, and DC measurements themselves, as the measurements increased with the passage of time and by continuously filming and recording five trials of each subject; thus, a bias would arise in the average values of the measurement pairs on the X-axis of the graph. In addition, since each subject underwent five repeated measurements, subject-specific errors occurred; thus, we avoided making comparisons of all 27 measurements trials.

Figure 6A is the graph of TI. The average difference of TI was −0.01 s, and the plot showed both a positive and negative distribution around the 0 point of the Y-axis. Statistically, there were neither fixed errors nor proportional errors. The time of soft palate rising movements (equivalent to VI) reported in previous studies ranged from 0.32 [32] to 0.5 s [33], which is consistent with the results of our study. Therefore, with regard to VI, we believe that the SI of the sensor can reflect the time from the soft palate being in the lowest and most retracted position (VA) to the highest and most advanced position (VB).

Figure 6B is the graph of TII. The average difference of TII was −0.33 s, and the plot shows a downward distribution from the center of Y-axis. Statistically, there was no proportional error, but a fixed error was determined. The average values were 0.68 ± 0.13 s for VII and 0.35 ± 0.15 s for SII. Figure 6C is the graph of TIII. The average difference of TIII was −0.34 s, and like TII, the plot showed a downward distribution from the center of Y-axis. Similarly, there was no proportional error, but a fixed error was noted. The average values were 1.06 ± 0.19 s for VIII and 0.73 ± 0.18 s for SIII. The SC did not match the VC because the TI was consistent, and the TII and TIII presented a fixed error. In addition, there was a time lag at regular intervals, which may have been reflected another organ or another soft palate movement. As previous studies have indicated that on average, the time taken from the soft palate to start rising to return to its original position was 1.159 s [32], we believe that the VIII in the results of this study was correctly measured. In addition, the opening of the Eustachian tube should occur at almost the same time that the soft palate reaches the highest position or 0.03–0.06 s later [32]. Since the sensor waveform reflects the movement of the ear canal, including the eardrum, it is possible that the SC did not match the VC due to the effect of the Eustachian tube opening and the middle ear pressure being balanced. Conversely, SII required about 0.35 s, and for all subjects, none matched the opening time of the Eustachian tube in the previous study, and it did not represent the opening of the Eustachian tube itself. In order to correspond to the point (VC), where the soft palate re-descended, it was necessary to reconsider the method of adopting SC points.

Regarding the sensor waveforms, SA, SB, and SC could be identified 27 times during 30 swallows, but there were cases where it was difficult to identify the point due to individual differences in the waveform. Figure 7 shows an example of waveforms for each subject. Figure 7A, showing the data of subject No. 3, presented less error than other subjects in all items from DA to DC. Figure 7B shows the data of subject No. 6. While the waveform of subject No. 3 was relatively simple and smooth, that of subject No. 6 showed small fluctuations. Similar trends were observed in subjects No. 2 and No. 5. Regarding this difference in waveform, a previous study pointed out that when measuring the pressure in the Eustachian tube, the carotid artery beats could be included in the pressure waveform because the carotid artery runs just behind the Eustachian tube [34,35]. Even in the case of an earphone-type sensor, it is possible that the waveform is pulsatile depending on the condition of the subject, such as the position of the sensor insertion, hypertension, and mental tension, and the point to be adopted may have been incorrect. It is necessary to reconsider which point to adopt when such a pulsatile waveform is observed. Subject No. 1 (Figure 7C) seemed to achieve a pulsatile waveform at first glance, but the error was the second smallest in subject No. 3. This was because the subject himself/herself pressed the switch before and after swallowing, and the interval was converted to a waveform based on it being defined as swallowing; thus, the difference in recognition of each subject was reflected in the length of time on the X-axis. Subject No. 1 pressed the switch at approximately 1.38 s, which was as shorter amount of time than for other subjects. Therefore, we believe that the waveform was stretched horizontally, and the position of the plot became clear. As a healthy person requires 0.5–1.0 s for the swallowing reflex, the difference in swallowing time observed between subjects here may have been due to the subjective view of each subject and the speed of pressing the switch. If the extraction time was too short or too long, there was a possibility of misunderstanding the points to be adopted; in the future, it will be necessary to devise methods able to extract the characteristics of the waveform using machine learning for a certain period of time before and after the swallowing reflex.

There were no fixed errors or proportional errors between VI and SI, which confirmed that the sensor measurements would be appropriate for clinical use. This represented the time taken for the soft palate to go from the lowest and most retracted position immediately before swallowing to the highest and most advanced position during swallowing. To date there have not been any methods or devices developed for conducting simple determinations of the timing and/or duration of soft palate movement. Consequently, to the best of our knowledge, there is no literature that provides objective values on the degree to which soft palate movement is delayed or shortened by pathology. Soft palate movement involves the tensor veli palatine muscle, the levator veli palatine muscle, the palatoglossus muscle, the palatopharyngeal muscle, the superior pharyngeal constrictor muscle, and the palatal ptosis muscle [36]. Because atrophy, weakness, rigidity, and ataxia of these muscles affect soft palate movement, the evaluation of VI (SI) can provide objective, clinically important information.

The first piece of information that can be obtained from the duration of swallowing is the time required to elevate the soft palate. While we were unable to confirm this with data from subjects other than healthy adults, clinically speaking, if the soft palate is paralyzed or atrophied it can be observed to elevate slowly or insufficiently and lower prematurely. If the timing of these events is off, the nasopharyngeal cavity may not close completely, potentially causing aspiration or reflux into the nasal cavity. Further, if the time from final chewing to the elevation of the soft palate (swallowing) is known, the time required for food bolus transport and the oral phase can be determined.

Regarding the compatibility between the VF and sensor measurements, approximately 63% of SI, approximately 15% of SII, and approximately 40% of SIII had relative errors less than ±30% and did not reach the 75% standard for compatibility. Therefore, the results indicated that the sensor and the VF were not compatible. However, the tolerance for concordance in the Bland–Altman analysis was not clearly defined and left to the discretion of the reporter. Here, it was determined that the number of measurements with a relative error less than ±30% was 75% or more, but re-examination should be considered based on reports in the same research field.

It is desirable to evaluate swallowing function in a series of steps from taking food into the oral cavity to chewing, feeding, and swallowing as much as possible; however, thus far, in order to evaluate in a series of flows, it was necessary to test under nonregular environments, such as radiation exposure and invasiveness. Since the earphone-type sensor can take measurements by simply attaching the earphone to the outer ear, it can evaluate in a normal eating environment without requiring a specific examination room. In addition, this measuring device does not interfere with the swallowing operation. Furthermore, there is no risk associated with pain during examination, radiation exposure, or the use of contrast media, which allows a simple and non-invasive evaluation to be performed. Because the device is small and lightweight, it can be used not only in hospitals but also in residences for older adults, home-based medical care, and nursing care facilities. Furthermore, it has been verified that the earphone-type sensor can measure mastication and breathing rate [20,21,22,23,24]. The combined use of earphone-type sensors and other non-invasive devices may enable continuous evaluation of swallowing function. For example, the pharyngeal phase of the swallowing process begins with a retraction of the soft palate leading to contact with the posterior pharyngeal wall. It follows that if the earphone-type sensor can detect the time from final chewing to SA, then the time required for Stage II transport [37] (the transportation of a food bolus) can also be measured. When a fixed amount of sample or foodstuff is taken into the oral cavity, the time required to chew, form a food bolus, and make it ready for swallowing (length of oral phase) can be measured as well. Ordinarily, swallowing begins during the expiratory phase of respiration, and after swallowing, respiration resumes with continued expiration [38]. We believe it is possible to evaluate the synchrony between swallowing and respiration by checking the phase of respiration at which swallowing has occurred as well as the appearance of the soft palate movement waveforms (SA and SB).

In future studies, it is necessary to reconsider the SC corresponding to the point where the soft palate descends (VC) and the method of adopting points in cases of a pulsatile waveform. Here, the analysis was based on a single subject, and thus it is necessary to confirm the inter-rater and intra-rater reproducibility. Additionally, it is necessary to ascertain whether similar results can be obtained with a larger cohort of subjects. We plan to conduct the same study for older individuals and patients with dysphagia. Furthermore, a method that can continuously evaluate the entire swallowing process in combination with the already verified measurements of mastication [20] and respiration [21] should be developed.

This study has several limitations. First, the sensor obtained waveforms from the ear canal, which includes the tympanic membrane, making it difficult to completely capture the sole movement of the specified organ. Second, the study did not take into account the possible influence of the presence of cerumen or other mitigating conditions within the ear canal on the resulting measurements. Third, since the strength of the waveform changes depending on the position and angle of the sensor, it is difficult to evaluate the muscle strength of each organ and its movement distance. Fourth, nasopharyngeal closure is said to be affected by posture [39], but in this study, measurements were performed in a sitting position with the head and neck trunk held in the intermediate position. Fifth, the presence or absence of aspiration cannot be confirmed. Sixth, although soft palate movement is known to be affected by the respiratory phase [40], this study did not take such influence into account. Finally, the results of this study are based on data from only six healthy individuals; thus, future studies will be needed to determine whether the same measurements and results can be obtained with larger cohorts. Furthermore, because of the limited sample, this study was unable to confirm the generalizability of this method for elderly and ill patients, which is our topic for future research.

## 5. Conclusions

To the best of our knowledge, this was the first attempt to evaluate swallowing function using a non-invasive approach from the ear canal. Using the earphone-type sensor, we were able to measure the time taken for the soft palate to move from its lowest and most retracted position before swallowing to the highest and most advanced position during swallowing. We were able to confirm the validity of its clinical application. Furthermore, the earphone-type sensor has not reached the level of compatibility with VF. To date, soft palate movement has been difficult to evaluate noninvasively, but the use of earphone-type sensors may provide important insights into the soft palate. In the future, we will aim to achieve the continuous evaluation of the swallowing processes, including mastication, bolus movement, and synchronization with respiration.

## Figures and Tables

**Figure 1 sensors-22-05176-f001:**
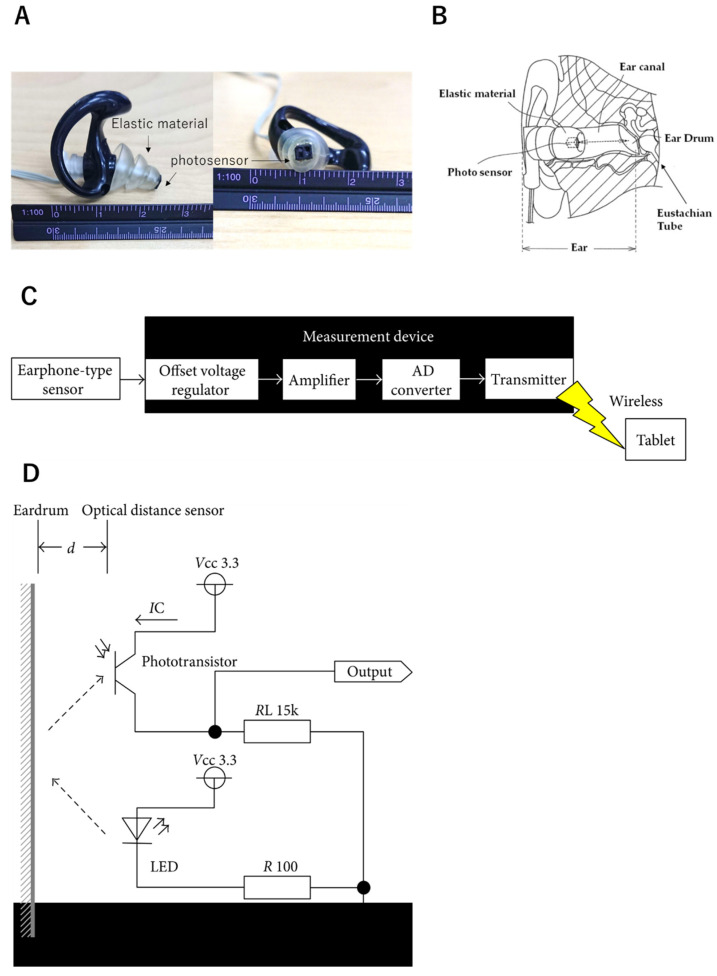
Earphone-type sensor. (**A**) Shape of the sensor used in this study. Elongation of the elastic material at the tip improved the sensor and allowed it to reach the eardrum. Since the elastic region models its shape to fit the shape of the ear canal, the sensor can accommodate for the individual differences in the shape of the ear canal. (**B**) The mechanism and structure of the earphone-type sensor. A small optical sensor is mounted on the tip of the sensor, and a light-emitting diode (LED) and a phototransistor are incorporated. By irradiating the ear canal with infrared light using an LED (wavelength 960 nm) and moving the irradiated area, changes in the reflected light emitted can be detected. When the phototransistor captures the reflected light signals, it detects the movement of the ear canal, including the eardrum. The light energy of the detected light is converted and amplified into electrical energy, and the signal is converted into a digital signal using an analog–digital (AD) conversion circuit (10 bits, 250 Hz). The results of these measurements can be captured and are recorded on a tablet terminal at a remote location using Bluetooth. (**C**) Configuration of the measurement device. (**D**) Electronic circuits surrounding the lightwave distance sensor.

**Figure 2 sensors-22-05176-f002:**
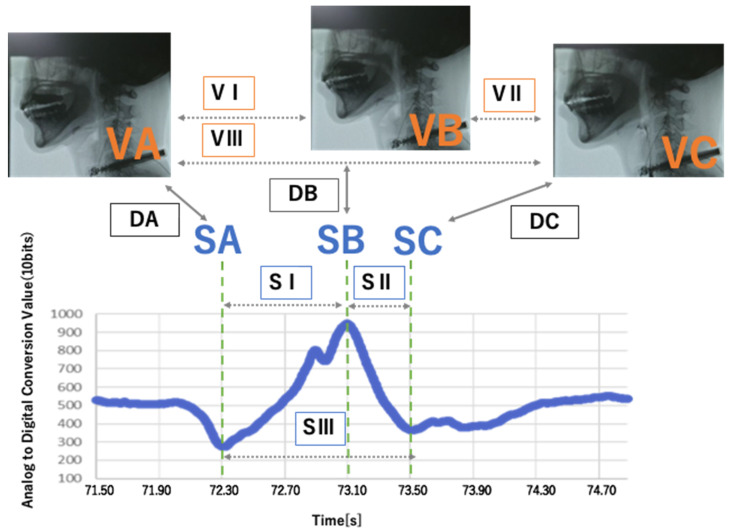
Correspondence between soft palate movement and sensor waveform.

**Figure 3 sensors-22-05176-f003:**
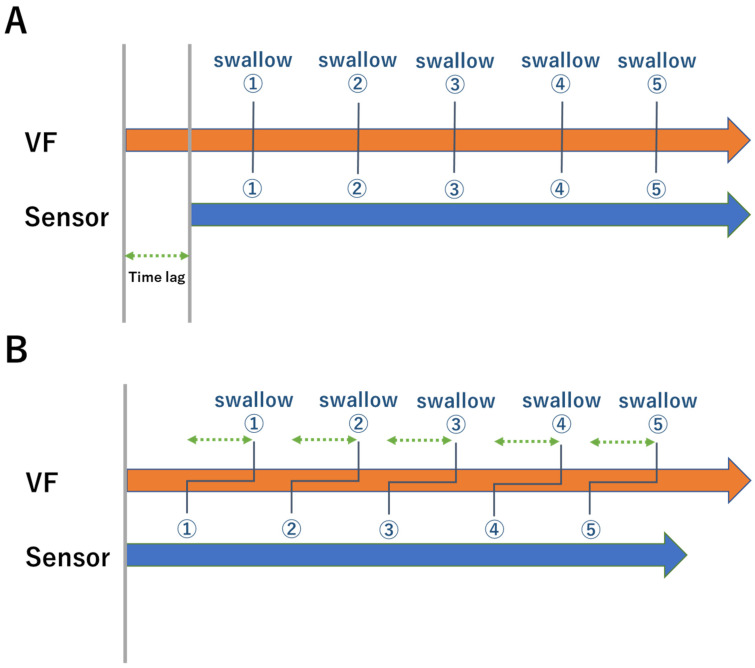
Data correction method. (**A**) Data from five swallowing trials were continuously recorded. As the start time of recording was different between the VF and the sensor, there a time lag in the data was recorded. (**B**) The time lag was determined on the basis of the audio recorded at the same time, and the corrected time of emergence of SA to SC was used for the analysis. Numbers in the figure show the number of swallowing trials. VF: Videofluoroscopy; SA: Time the waveform was at its lowest point; SC: Time the waveform decreased.

**Figure 4 sensors-22-05176-f004:**
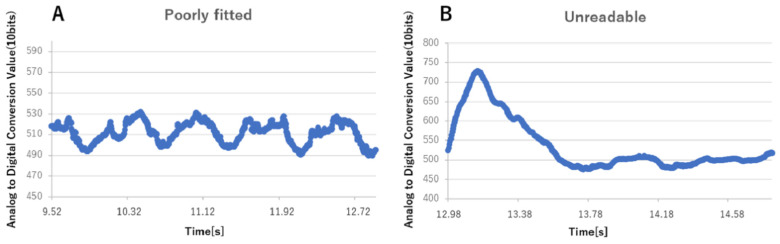
An example of a waveform left out of analysis. (**A**) Poor sensor mounting (subject No. 2), (**B**) Indiscernible data (subject No. 4).

**Figure 5 sensors-22-05176-f005:**
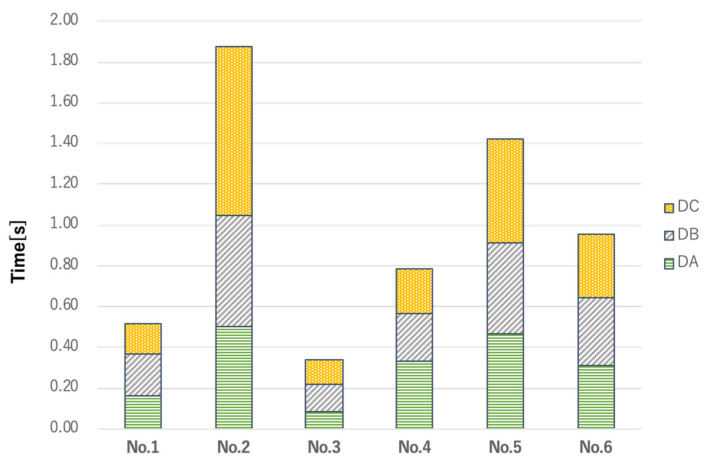
Sum of errors in sensor measurements vs VF measurements. The sum of sensor error (DA + DB + DC) versus VF for each subject is graphed from the results in Table 1. The smallest errors in the VF and sensor readings were observed in No. 3, with errors of DA: 0.09 ± 0.07 s, DB: 0.13 ± 0.07 s and DC: 0.12 ± 0.04 s. The largest errors were seen for subject No. 2, with DA: 0.50 ± 0.15 s, DB: 0.54 ± 0.12 s and DC: 0.83 ± 0.30 s. DA: Difference in the time of emergence of VA and SA; DB: Difference in the time of emergence between VB and SB; DC: Difference in the time of emergence of VC between SC; VF: Videofluroscopy.

**Figure 6 sensors-22-05176-f006:**
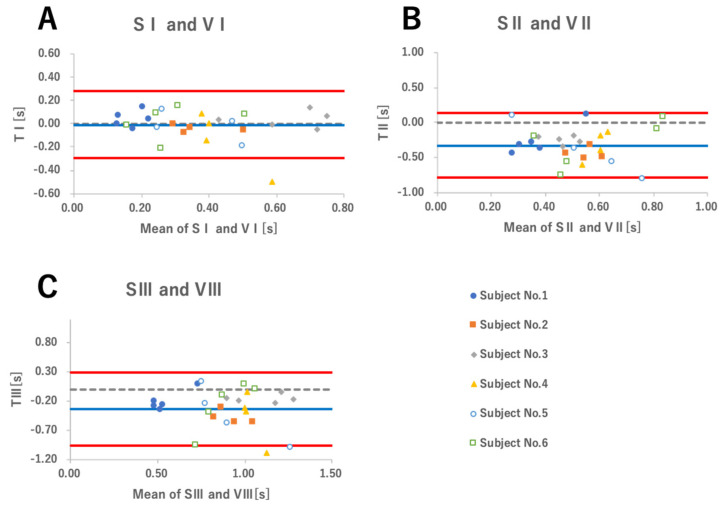
Bland–Altman plot. The X-axis shows the average value (seconds) of the measurement pairs and the Y-axis shows the difference (seconds) between the methods. The blue line on the graph shows the average value of the difference between the two measurement methods, and the red line shows the limits of agreement. (**A**) VI and SI. TI had a mean difference of −0.01 ± 0.14 s. Neither fixed nor proportional error was observed. (**B**) VII and SII. TII had a mean difference of −0.33 ± 0.23 s. There was a fixed error, but no proportional error was observed. (**C**) VIII and SIII. TIII had a mean difference of −0.34 ± 0.31 s. There was a fixed error, but no proportional error was observed. VI: VI time from VA to VB; VII: time from VB to VC; VIII: time from VA to VC; SI: time from SA to SB; SI: SI time from SA to SB; TI: difference between VI and SI; VII: Time from VB to VC; SII: Time from SB to SC; VIII: Time from VA to VC; SIII: Time from SA to SC; TIII: VIII and SIII.

**Figure 7 sensors-22-05176-f007:**
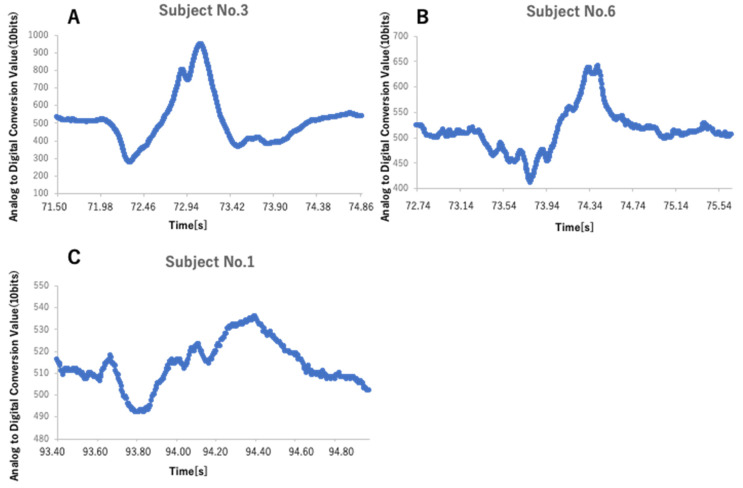
Example of waveforms for each subject. (**A**) Waveform of Subject No. 3, which is simpler and smoother than those of No. 6 and 7. Subject No. 3 had fewer errors than the other subjects on all items from DA to DC. (**B**) Waveform of Subject No. 6. It shows small tremors, suggesting the possibility that the carotid pulse was also captured by the recording. (**C**) Waveform of Subject No. 1, which also shows small tremors as in B. However, it is possible that the subject was only aware of swallowing for a short time, which caused the waveform to be stretched horizontally, thus clarifying the position of the plot. DA: Difference in the time of emergence of VA and SA; DC: Difference in the time of emergence of VC between SC.

**Table 1 sensors-22-05176-t001:** Results of the measurements for each subject.

Subject	DA	DB	DC
	Mean ± SD	Mean ± SD	Mean ± SD
No. 1	0.16 ± 0.02	0.20 ± 0.08	0.15 ± 0.05
No. 2	0.50 ± 0.15	0.54 ± 0.12	0.83 ± 0.30
No. 3	0.09 ± 0.07	0.13 ± 0.07	0.12 ± 0.04
No. 4	0.33 ± 0.16	0.23 ± 0.13	0.22 ± 0.30
No. 5	0.47 ± 0.35	0.44 ± 0.36	0.51 ± 0.42
No. 6	0.31 ± 0.10	0.33 ± 0.18	0.31 ± 0.19
Weighted average	0.30 ± 0.21	0.30 ± 0.21	0.34 ± 0.33

Unit [s]. DA: Difference in the time of emergence of VA and SA; DB: Difference in the time of emergence between VB and SB; DC: Difference in the time of emergence of VC between SC; SD: Standard Deviation.

**Table 2 sensors-22-05176-t002:** Results of the measurements for each subject.

Subject	VI	SI	VII	SII	VIII	SIII
	Mean ± SD	Mean ± SD	Mean ± SD	Mean ± SD	Mean ± SD	Mean ± SD
No. 1	0.15 ± 0.05	0.19 ± 0.06	0.51 ± 0.04	0.24 ± 0.21	0.66 ± 0.04	0.43 ± 0.19
No. 2	0.39 ± 0.10	0.34 ± 0.09	0.77 ± 0.07	0.32 ± 0.07	1.17 ± 0.14	0.67 ± 0.08
No. 3	0.60 ± 0.16	0.67 ± 0.12	0.59 ± 0.07	0.34 ± 0.06	1.19 ± 0.17	1.01 ± 0.17
No. 4	0.51 ± 0.22	0.37 ± 0.05	0.76 ± 0.07	0.43 ± 0.14	1.27 ± 0.28	0.80 ± 0.17
No. 5	0.38 ± 0.18	0.36 ± 0.10	0.76 ± 0.40	0.34 ± 0.02	1.14 ± 0.46	0.70 ± 0.10
No. 6	0.29 ± 0.13	0.31 ± 0.17	0.75 ± 0.16	0.44 ± 0.36	1.03 ± 0.11	0.74 ± 0.34
Weighted average	0.38 ± 0.20	0.38 ± 0.18	0.68 ± 0.19	0.35 ± 0.19	1.06 ± 0.29	0.72 ± 0.26

Unit [s]. VI: Time from VA to VB; VII: Time from VB to VC; VIII: Time from VA to VC; SI: Time from SA to SB; SII: Time from SB to SC; SIII: Time from SA to SC; SD: Standard Deviation.

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
