# Peer review of "Validation of Earphone-Type Sensors for Non-Invasive and Objective Swallowing Function Assessment"

_sensors, 2022, doi:10.3390/s22145176_

Round 1

Reviewer 1 Report

The authors present validation of earphone-type sensor for non-invasive and objective swallowing function assessment. It is a very interesting manuscript with a novel research idea. The research is well conducted and the results are very clear and concise. I want the authors to consider the following comments for publication.

1- What is the plastic material used to house the sensor? Is it a commercially available product or specially designed for the said purpose? Provide the details in the manuscript.

2- The authors have provided the schematic representation of the sensing device. It would be great if the authors also explain the details of the overall sensing mechanism involving photo emitters and photodetectors and how the response was transslated?

3- Although the details are explained but there is no mention of Figure 2 in section 2.3? 

4- Figure captions should be self descriptive explaining all the components of the Figures. All the Figure captions need revision.

5- The mechanism and the shape of the sensor should be explained later in the methods or results rather than in the introduction.

6- It would be great if the authors provide a brief comparison of their device with the similar non-invasive device commercially available in the market?

Author Response

Response to Reviewer 1 Comments

1- What is the plastic material used to house the sensor? Is it a commercially available product or specially designed for the said purpose? Provide the details in the manuscript.

Thank you for pointing out this omission. The elastic material that houses the sensor was prepared by processing a commercially available material. I have added the following to the manuscript in response to your comment:

"To create the elastic material, the commercially available earplugs were altered  (EP4 SONIC DEFENDERS: PLUS Filtered Flanged Earplugs (SureFire LLC., Fountain Valley, CA, USA)."

2- The authors have provided the schematic representation of the sensing device. It would be great if the authors also explain the details of the overall sensing mechanism involving photo emitters and photodetectors and how the response was translated?

Thank you for your guidance on this. We have added schematics and descriptions to the manuscript as Figures 1C and 1D (shown below).

The earphone-type sensor was the same shape as earplug-style earphones. Its built-in optical distance sensor “QRE1113” (Fairchild Semiconductor International Inc.) used an infrared LED and phototransistor to penetrate the inner ear canal with infrared light. The reflected light was then captured by the phototransistor. The movement of the ear canal during swallowing could be measured using the data thereby obtained (Fig. 1C). Figure 1D shows the electronic circuits surrounding the lightwave distance sensor. As Fig. 1D depicts, when the distance (d) between the tympanic membrane and the optical distance sensor decreased, the amount of light reflected from the tympanic membrane increased, along with the output voltage. Likewise, an increase in distance (d) led to a decrease in reflected light and output voltage.

The earphone-type sensor was connected via cable to the measuring device. The measuring device was 110x75x25mm and weighed 115g; it was small enough that it would be able to sit on a dining table without getting in the way. The measuring device supplied a voltage of DC3.3V to the earphone-type sensor, and the sensor's output data was detected by the measuring device’s offset voltage regulator. The offset voltage levels measured from the sensor were adjusted to central values after AD (analog-digital) conversion of the signal received by the offset voltage regulator of the measuring device. The adjustment was then set to the center value of the AD convertible range = 3.3 V (power supply voltage of the AD converter) / 2 = 1.65 V. This adjustment of the offset voltage was necessary to compensate for differences in this parameter caused by individual differences in the shape of the ear canal. The value (waveform) measured by the sensor was the amplitude based on the offset voltage. Because the offset voltage was fixed by the amplifier, only the amplitude was enlarged in the signal after adjustment of offset voltage. Using a knob (variable resistor) on the instrument, the amplification level could be raised up by to 40x. After amplification, the analog signal was converted to a digital signal by an AD converter with a sampling frequency of 250 Hz and a resolution of 10 bits. Finally, the converted digital signal was transmitted to a tablet (ASUS Nexus 7, Bluetooth 3.0) by a transmitter (Bluetooth 2.1).

3- Although the details are explained but there is no mention of Figure 2 in section 2.3? 

Thanks for pointing that out. We have added Figure 2 to Section 2.3 of the manuscript.

4- Figure captions should be self descriptive explaining all the components of the Figures. All the Figure captions need revision.

Thank you for your guidance on this issue. We have added descriptive explanations to the footnotes of Figures 2, 5, 6, and 7.

5- The mechanism and the shape of the sensor should be explained later in the methods or results rather than in the introduction.

Thank you for your advice. We have accordingly moved the description of the sensor’s shape from the introduction to the methods section.

6- It would be great if the authors provide a brief comparison of their device with the similar non-invasive device commercially available in the market?

Thank you for your helpful suggestion. As far as we were able to tell, there are no devices currently on the market that are capable of non-invasively assessing soft palate movements, the subject of investigation in this study. Therefore, we were unable to make a comparison. However, we have added a brief note stating this in the Discussion on page 22.

Reviewer 2 Report

This is an interesting clinical report.

I have minor comments:

You need to correct the references format in the paper.

I would like to get response to the following questions, which are because of my ignorance in the subject:

1/ I wonder if wax at the ears may affect the results.

2/ I wonder how much information can be obtained from the duration of the swallowing. 

3/ Is the small number of tested patients adequate

4/ Could we see the difference between healthy and sick test?

5/ How did the Helsinki committee approve the use of the non-standard equipment (sensor)?

Perhaps the above questions may interest other readers.

I wish you much success in your work

Author Response

Response to Reviewer 2 Comments

1/ I wonder if wax at the ears may affect the results.

Thank you for your helpful suggestion. As you correctly point out, it is impossible to deny that earwax could affect the measurements. Because we were unable to take this into account in this study, we have added a line in the Limitations section to reflect that.

2/ I wonder how much information can be obtained from the duration of the swallowing.

Thank you for your question. The first piece of information that can be obtained from the duration of swallowing is the time required to elevate the soft palate. While we were unable to confirm this with data from subjects other than healthy adults, clinically speaking, if the soft palate is paralyzed or atrophied it can be observed to elevate slowly or insufficiently and lower prematurely. If the timing of these events is off, the nasopharyngeal cavity may not close completely, potentially causing aspiration or reflux into the nasal cavity. Further, if the time from final chewing to the elevation of the soft palate (swallowing) is known, the time required for food bolus transport and the oral phase can be determined. We have added passages explaining this to the manuscript.  

3/ Is the small number of tested patients adequate

Thanks for pointing that out. Since we only had six subjects for this study, we believe future studies will be needed to determine whether the same measurements and results can be obtained with larger cohorts. We have added passages addressing this to the limitations section of the Discussion in the manuscript.

4/ Could we see the difference between healthy and sick test?

Because selection criteria for this study were limited to healthy adults between the ages of 20 and 60, we were unable to confirm whether the measurements can be made in the elderly and infirm. We have added a note acknowledging this to the Limitations section of the manuscript.

5/ How did the Helsinki committee approve the use of the non-standard equipment (sensor)?

Thanks for raising this issue. The sensor used in this study was developed as a device for measuring mastication count per unit time. The Ethics Review Committee approved this study because it was non-invasive and designed to confirm the reproducibility of comparisons with VF. The research methods were explained to the subjects using a document, and consent was obtained in writing.

Round 2

Reviewer 1 Report

The revised manuscript looks in good shape. I believe it would be a good publication for sensors. Hence, I would like to accept the revised manuscript in the present form.